# MR-LINAC, a New Partner in Radiation Oncology: Current Landscape

**DOI:** 10.3390/cancers16020270

**Published:** 2024-01-08

**Authors:** Abrahams Ocanto, Lisselott Torres, Miguel Montijano, Diego Rincón, Castalia Fernández, Beatriz Sevilla, Daniela Gonsalves, Macarena Teja, Marcos Guijarro, Luis Glaría, Raúl Hernánz, Juan Zafra-Martin, Noelia Sanmamed, Amar Kishan, Filippo Alongi, Drew Moghanaki, Himanshu Nagar, Felipe Couñago

**Affiliations:** 1Department of Radiation Oncology, Hospital Universitario San Francisco de Asís, GenesisCare, 28002 Madrid, Spain; lisselott.torres@genesiscare.es (L.T.); miguel.montijano@genesiscare.es (M.M.); diego.rincon@genesiscare.es (D.R.); castalia.fernandez@genesiscare.es (C.F.); beatriz.sevilla@genesiscare.es (B.S.); daniela.gonsalves@genesiscare.es (D.G.); macarena.teja@genesiscare.es (M.T.); marcos.guijarro@genesiscare.es (M.G.); alberto.glaria@genesiscare.es (L.G.); raul.hernanz@genesiscare.es (R.H.); felipe.counago@genesiscare.es (F.C.); 2Department of Radiation Oncology, Hospital Universitario Vithas La Milagrosa, GenesisCare, 28010 Madrid, Spain; 3Group of Translational Research in Cancer Immunotherapy, Centro de Investigaciones Médico-Sanitarias (CIMES), Instituto de Investigación Biomédica de Málaga (IBIMA), Universidad de Málaga (UMA), 29010 Málaga, Spain; jzafra08@gmail.com; 4Department of Radiation Oncology, Hospital Universitario Virgen de la Victoria, 29010 Málaga, Spain; 5Department of Radiation Oncology, Hospital Universitario Clínico San Carlos, 28040 Madrid, Spain; noelia.sanmamed@salud.madrid.org; 6Department of Radiation Oncology, University of California, Los Angeles, CA 90095, USA; aukishan@mednet.ucla.edu; 7Advanced Radiation Oncology Department, Cancer Care Center, IRCCS Sacro Cuore Don Calabria Hospital, 37024 Negrar, Italy; filippoalongi@libero.it; 8University of Brescia, 25121 Brescia, Italy; 9UCLA Department of Radiation Oncology, University of California Los Angeles, Los Angeles, CA 90095, USA; dmoghanaki@mednet.ucla.edu; 10Department of Radiation Oncology, Weill Cornell Medicine, New York, NY 10065, USA; hnagar@med.cornell.edu; 11GenesisCare, 28043 Madrid, Spain

**Keywords:** adaptative radiotherapy, image guidance, stereotactic body radiotherapy, radiotherapy

## Abstract

**Simple Summary:**

This manuscript is entitled “MR-LINAC, a new partner in radiation oncology: current landscape” in response to your invitation to collaborate on the Special Issue on “Hypofractionated Radiotherapy in Cancer Treatments”. The main contribution of this review is to provide an update on the scientific evidence regarding the role of new technology in radiation oncology: the MR-LINAC. These data are relevant because there have been important advances in the management of different pathologies with this LINAC, principally high dose and SBRT, with promising results in recent years.

**Abstract:**

Technological advances in radiation oncology are oriented towards improving treatment precision and tumor control. Among these advances, magnetic-resonance-image-guided radiation therapy (MRgRT) stands out, with technological advances to deliver targeted treatments adapted to a tumor’s anatomy on the day while minimizing incidental exposure to organs at risk, offering an unprecedented therapeutic advantage compared to X-ray-based IGRT delivery systems. This new technology changes the traditional workflow in radiation oncology and requires an evolution in team coordination to administer more precise treatments. Once implemented, it paves the way for newer indication for radiation therapy to safely deliver higher doses than ever before, with better preservation of healthy tissues to optimize patient outcomes. In this narrative review, we assess the technical aspects of the novel linear accelerators that can deliver MRgRT and summarize the available published experience to date, focusing on oncological results and future challenges.

## 1. Introduction

Radiation therapy is a crucial part of cancer treatments where a multi-disciplinary approach is the standard of care. The advances in radiation oncology are continuous, due to innovations in engineering, imaging, physics, and technology, along with an improvement in the knowledge of the radiobiology of cancer and healthy tissues. The technology used in radiation therapy (RT) and other oncological treatments is constantly evolving. In less than 40 years, we have moved from two-dimensional radiotherapy (2DRT) to three-dimensional RT (3DRT) with the invention of computed tomography (CT) in 1972 [1]. Further technological advances led to the development of inverse computational algorithms, leading to the introduction of intensity-modulated RT (IMRT), which further improved dose conformation, allowing for decreased toxicities while allowing for safe dose escalation studies [2]. Further advances have been achieved with volumetric modulated arc therapy and the integration of on-board cone-beam CT (CBCT) to improve image-guided radiation therapy precision (IGRT) [2].

The availability of on-board CBCT imaging facilitated the introduction of adaptative RT (ART), which aims to surpass the benefits of IGRT alone. This concept was introduced in 1997 [3] and consists of adapting the delivery of RT to the structural and positional changes in the tumor and organs at risk in between treatments. A persistent limitation of ART regards the quality of CBCT images, particularly in the thorax and abdomen. This creates difficulties in distinguishing the boundaries between tissues, introducing uncertainties and patient safety concerns, limiting dose escalation opportunities. These concerns partly explain why ART has not been adopted more broadly in radiation oncology. Further concerns include an inability to visualize the target and avoidance structures while radiation therapy is being delivered.

In contrast, magnetic resonance imaging, discovered in 1973 by Lauterbur [4], often provides better differentiation between a tumor and healthy tissues and can detect subtle physiological changes not appreciable with CBCT. This technology has been integrated into lineal accelerators (LINACs), referred to as MR-LINACs, and can provide real-time images during treatment to deliver what is commonly referred to as MRI-guided radiation therapy (MRgRT). With this new technology, ART has become safer through enhanced visualization of internal structures and quality control given the ability to monitor anatomic structures and positions while treatments are delivered. This has also improved the safety of stereotactic body radiotherapy (SBRT) or stereotactic ablative radiotherapy (SABR), designed to deliver the highest doses commonly prescribed by radiation oncologists [5]. Without MRgRT technologies, SBRT or SABR treatment accuracy must rely on X-ray-based images taken before, but not during, treatment [6].

For this review, we will discuss the advantages of MR-LINACs compared to conventional LINACs: improved organ delineation compared to CT, the opportunity for real-time ART with the capability to monitor the target during treatment (gating), reduced margins during treatment planning, and others [7,8,9]. This review focuses on the characteristics of this new technology in radiation oncology, its use for the treatment of different types of solid tumors, as well and its benefits, limitations, and future perspectives.

## 2. Characteristics of MR-LINACs

The integration of the MRI imaging capability with a modern linear accelerator represents a paradigm shift in radiation therapy due to the opportunity to use the real-time anatomic and physiologic changes in the tumor and in the surrounding critical organs to optimize the therapeutic ratio.

The introduction of MR-LINACs has brought up again the issue of tumor tracking through gating. This technique allows for reduced uncertainties during treatment delivery due to the movement of critical structures and the tumor. Not only because of breathing, but also intestinal peristalsis, bladder filling, etc. Historically, this problem has been accounted for through the establishment of wide planning margins, which in turn caused more healthy-tissue irradiation and, therefore, higher toxicity [10]. There are currently four MR-LINAC systems: two of them installed around the world (Unity by Elekta (Stockholm, Sweden) and MRIdian by ViewRay (Ohio, OH, USA)) and the other two in the development stage. The designs vary in their electromagnetic field and image quality (with some systems having high fields with better image quality and others having low fields with worse quality but less electromagnetic interactions) as well as dosimetric effects [3].

Of these four MR-LINACs, the one with the highest-intensity electromagnetic field is Elekta’s Unity, designed in The Netherlands/Utrecht and based on a 1.5 T magnet (Philips, Ingenia, The Netherlands, Eindhoven) with a small gap in the gradient coil and the magnet winding [11]. The Australian prototype, currently being studied in a phase II clinical trial, has a design with an open 1 T magnet that can direct the beam in any direction, which makes it a versatile unit for image obtention and treatment [12]. Aurora-RT (MagnetTx Oncology Solutions, Edmonton Alberta/Canada) is equipped with a 0.5 T magnet [13]. MRIdian has a 0.35 T split magnet that has an adequate imaging quality and has the great advantage of minimizing the interactions with the magnetic field compared to other systems [14].

The two currently installed MR-LINACs have differences in their design and treatment characteristics, as mentioned above. However, both systems use coplanar static IMRT fields. The dose rate is 650 MU/min for MRIdian and 500 MU/min for Unity. The gantry rotation speeds are 0.5 and 6 rpm, respectively, and both have no collimator rotation [10,15].

With respect to image acquisition, MRIdian uses a steady-state free precession (SSFP) sequence for treatment planning and delivery, with a very high resolution. It is also possible to acquire T1, T2, and DWI sequences to be used in conjunction with the SSFP image. Unity provides a gamut of pulse sequences for planning and treatment. Images in both systems are used for gating during treatment, with the gating in MRIdian acquiring several cutplanes (sagittal, coronal, and axial) or in a single cutplane at 8 frames per second. The radiation beam stops automatically when the volume exceeds the defined margins. The Unity system can perform tracking in three planes (sagittal, coronal, and axial) [10]. Recently, Unity received FDA approval for the Comprehensive Motion Management tool (CMM), already implemented in some centers and soon to be widely available. While anatomy-specific imaging sequences, encompassing T1, T2, Balanced, FLAIR, SPAIR, and 3D Vane, are utilized to optimize target and healthy-tissue visualization, CMM involves the use of balanced contrast imaging for respiratory targets and slower T2 cine for non-respiratory targets. Target definition is based on daily 3D MRI, with template images generated to address contrast discrepancies between cine and 3D images. Two essential motion management structures, the registration structure and gating envelope, are introduced. Four distinct motion management strategies—free-breathing exhale, free-breathing average, breath-hold, and an exception strategy—are delineated. Intrafraction tumor drifts may be corrected by means of the baseline shift (BLS), measuring the amount of drift and sending the data back to Monaco Online to adjust the segments to the current average tumor position and to recalculate in real time the updated dose and restart the treatment with the updated segment positions. The True Tracking and Automatic Gating system employs a predictive algorithm to compensate for system latency, continuously updating a motion model based on true tracking measurements. Distortion correction is applied to 2D cine images to enhance their accuracy. The integration of CMM routines in Elekta Unity enables continuous real-time tracking and precise treatment in the presence of motion.

Although MRI-guided radiation therapy was first pioneered by ViewRay with a cobalt delivery system, the first MR-LINAC was installed in May 2017. In that year, Elekta’s Unity in The Netherlands/Utrecht performed the first treatment in humans in a small cohort of patients with spinal metastases. In July of that same year, MRIdian treated the first patients in the Henry Ford Hospital in Detroit, USA, commencing a new era of MRgRT [11] (Table 1).

With MR-LINACs, ART is performed with the patient positioned in the machine, with two techniques developed by Elekta’s Unity: adapt-to-shape (ATS) and adapt-to-position (ATP). The first is a more complex process that requires recontouring and replanning while the patient is on the treatment table [9]. This is the more robust option but is also time-consuming. On the other hand, ATP consists of a virtual movement of the isocenter without the need for recontouring. This option is applied mainly in favorable anatomic sites where few interfraction changes exist [9] and helps optimize the treatment duration. An intermediate scenario is ATS-Lite, in which the target remains stable in each fraction and an adaption of relevant organs at risk (OARs) is performed. With modification in the workflow, one or two contours can be edited [9]. Among the advantages of daily-adapted plans, the option to evaluate the dose administered in each fraction and the way in which the daily plan can be adapted and further optimized (taking into account dose accumulation) has an impact on acute toxicity [9].

MR-LINAC has brought changes in the workflow of radiation oncology departments in terms of patient positioning, image obtention, recontouring, replanning, and the obtention of new images (MRI3D or cineMRI) to initiate gating and/or tracking [16]. The versatility of MR-LINAC makes it possible to treat several tumor sites, especially those where the contrast between tissues helps to better identify structures and guarantees a safe treatment. The most experienced centers, as well as ESTRO, recommend starting with localized prostate cancer, pancreatic cancer, liver tumors, and pelvic lymph nodes [16].

## 3. Clinical Applications of MR-LINAC

### 3.1. Prostate Cancer

MRI is widely implemented for the diagnosis and staging of prostate cancer [17], as it can differentiate the structures surrounding the gland (rectum, bladder, and intestine). It is also useful for image fusion to delineate RT treatment volumes. However, this provides just a static image before treatment that can differ during therapy. In this context, the daily adaptation with MR-LINAC can provide an advantage.

Several prospective studies of SBRT with MR-LINAC have been published in this scenario and it is a subject of growing interest. Kishan et al. [18] published the MIRAGE trial, which is the first phase 3 randomized trial to compare CT-guided SBRT against MRI-guided SBRT in patients with localized prostate cancer. Based on MRI imaging, a reduced 2 mm planning target volume (PTV) was compared to conventional margins in CT, and acute genitourinary (IPSS, EPIC-26) and gastrointestinal grade ≥ 2 toxicity up to 90 days post-treatment was assessed. They randomized 156 patients with a median age of 71 in both groups. Thirty-seven patients (24%) received pelvic irradiation and eighty-nine (44%) were treated with a rectal spacer. Moreover, 26% received an integrated boost to the dominant lesion and 68% were treated with hormonal therapy. In both groups, a simulation CT with 1.5 mm thickness with comfortable bladder filling and an empty rectum was performed. For patient contouring in the MR-LINAC group, a simulation MRI with TRUFI and T2 sequences was acquired. The PTV margins were 2 mm, and planning was performed with step-and-shot in MRIdian. During treatment, images were acquired in CINE and gating was established with 10% prostatic volume outside the margin. ATS was not performed, which could impact long-term toxicity. The CT group was planned with VMAT, 2–5 mm PTV, and CBCT prior to each fraction. The doses were 40 Gy in five fractions of 8 Gy, on consecutive or nonconsecutive days. The simultaneous boost was 42 Gy in five fractions. In patients with pelvic irradiation, the treatment was 35 Gy in five fractions. Genitourinary grade ≥ 2 toxicity was 22.4% in the MR-LINAC cohort compared to 43.4% in the CT group. A 10.5% decrease in acute gastrointestinal toxicity was observed in the MRI group. The percentage of patients with a significant increase in IPSS at one month was 6.8% in the MRI group cohort vs. 19.4% in the CT group, with no significant difference at 3 months. In patients treated with MRI, there was a decrease in EPIC-26 at one month compared to the CT group, but no differences were reported at 3 months. In short, MRI-guided SBRT in localized prostate cancer reduces genitourinary toxicity to 22.4% and gastrointestinal up to 0%. Based on forthcoming results in terms of disease control and survival, this could become a new gold standard in this setting. 

In June 2023, the first systematic review and meta-analysis comparing acute toxicity between these two techniques was published [19]. The reported acute genitourinary and gastrointestinal grade ≥ 2 toxicity rates were 16% and 4% vs. 28% and 9%, respectively. The gastrointestinal toxicity was slightly higher than in MIRAGE. This could be explained by the 2 mm margin and the 47% of patients with rectal spacers. The genitourinary toxicity was lower, probably because most studies use doses of less than 40 Gy, which generates less urethral toxicity. This further supports that ART considerably limits the dose to the bladder, rectum, and urethra, and has a significant clinical impact on toxicity [20]. More definitive oncological results are pending.

EXCALIBUR (NCT04915508) is a phase II non-randomized trial, the successor to SCIMITAR [21], that will exclusively recruit with MRIgRT and determine the influence on GI and GU reported outcomes. MRIgRT is also similarly being evaluated in the SHORTER trial (NCT04422132) [22], which will evaluate patients randomizing to either 36.5 Gy in 5 fractions or 55 Gy in 20 fractions on the MR-LINAC. The end point will be comparing the tolerability between these moderate and ultra-hypofractionated arms.

Another interesting topic in prostate cancer is the boost to the dominant prostatic lesion, which could be better visualized during ART. However, there no current studies that have followed this approach, save for a minority of patients in MIRAGE. To note, the phase 3 trial FLAME [23] compared standard external-beam radiotherapy in a conventional LINAC in intermediate- and high-risk tumors (77 Gy in 2.2 Gy per fraction) with an arm including a boost to the prostate lesion (94.5 Gy in 2.7 Gy per fraction). Five-year biochemical-recurrence-free survival was 92% in the experimental arm vs. 85% in the control arm, with this difference being statistically significant. Acute genitourinary grade ≥ 2 toxicity was higher in the boost group (28 vs. 23%), but this was not statistically significant. These results suggest a biochemical benefit of delivering a boost to the dominant lesion. MR-LINACs will allow for further studies in this scenario.

Boosting the visible intraprostatic macroscopic site of disease has been demonstrated to optimize biochemical-disease-free survival with no significant impact on genitourinary (GU) and gastrointestinal (GI) toxicity. HERMES is a single-center randomized trial in patients with intermediate–high-risk prostate cancer. Patients were randomized to receive 36.25 Gy in five fractions over 2 weeks or 24 Gy in two fractions over 8 days, with an integrated boost to the magnetic resonance imaging (MRI) visible tumor of 27 Gy in two fractions. Treatment was performed using the Unity MR-LINAC with a daily online adaptive approach. The interim analysis was promising, showing that GU toxicity in the two-fraction treatment was below the prespecified threshold (5/10 grade 2+) [24]

Thus MR- LINAC platforms, especially in prostate cancer, allow for a safe reduction in margins while maintaining target coverage (Figure 1), pushing the boundaries in hypo-fractionation and in the optimization of biological doses to the tumor. Thereafter, a series of randomized noninferiority trials comparing these extreme schedules using MR-guided approaches is warranted (Table 2).

### 3.2. Lung Cancer

Central lung tumors are thought to be the main candidates for MRgRT, as shown by Tekatli et al. [25]. While these treatments can be performed in a single fraction (as reported in SAFRON II), this approach does not offer significant differences in disease-free survival when compared to more fractions in oligometastatic patients [26]. ART in a single fraction takes more time; therefore, generally a “fractionated” treatment is chosen: during the same day, treatment replanning occurs twice (mainly due to patient discomfort).

Teklatli et al., in a prospective study, included 50 patients with primary lung tumors or lung metastases treated with single-fraction SBRT in MR-LINAC 0.35 T with doses between 20 and 34 Gy in central tumors and with a 5 mm PTV, step-and-shoot IMRT, 3 mm gating, and a sagittal CINE image. The two-year OS was 89% for patients with primary tumors and 63% for metastatic patients. Local failure occurred in two patients (4%) and median three-year local control was 97%, with only one patient developing grade 3 toxicity (chest wall pain). This shows that SBRT guided by MRI with reduced margins and respiratory gating achieves excellent local control and low toxicity [25].

An analysis of the treatment planning during breath-hold showed that PTV coverages of 95% or even higher can be achieved with this technique in central tumors with no high doses to OARs. However, its use has not been standardized yet for lung tumors [26].

Finazzi et al. assessed 25 patients with peripheral tumors treated with MR-LINAC in three to eight fractions with daily ART and breath-hold. The median treatment time was 48 min, with a PTV coverage of 92% in the predicted plan and 95% in the reoptimized plan (both with BED ≥ 100 Gy). The authors concluded that this treatment limits planning margins compared with the traditional ITV, although with a modest dosimetric benefit and with local control, toxicity, and survival results pending [27]. 

There is also an ongoing phase I trial of dose escalation for ultracentral tumors using SMART, delivering 55 Gy/10 fractions and up to 65 Gy/10 fractions (NCT04925583). An additional potential gain of MRgRT for dose escalation is the integration of functional imaging in a treatment adaptive approach. MR sequences can provide not only anatomical, but also functional, details that may be integrated as guidance for the adaptive strategies and to evaluate response to the treatment. For example, diffusion-weighted imaging (DWI) and apparent diffusion coefficients (ADCs) have been evaluated in improving tumor contouring, assessing treatment response, and defining the residual tumor [28]. An increase in the ADC value during therapy can be related to treatment-related cell death and it may be a potential biomarker to guide dose escalation for poorly responding tumors [29]. Dynamic contrast-enhanced (DCE) MRI is applied to assess tissue perfusion, blood flow, and tumor vasculature. Preliminary studies on several disease sites have suggested that increased enhancement during radiation treatment may be related to improved outcomes [30]. The use of MRgRT is novel and dose escalation using MR anatomical and functional imaging in the context of historically high-risk central lung tumors warrants prospective study.

Overall, MRgRT for thoracic tumor sites has shown promising results, with various investigations actively seeking to improve the therapeutic ratio in this setting.

### 3.3. Gastroenterological Tumors

RT delivery in the upper abdomen is limited by the radiosensitivity of OARs (duodenum, stomach, small, and large intestine), respiratory movement, peristalsis, and the difficulty in visualizing structures during planning and delivery in CBCT. For all the above, reaching ablative doses with SBRT and conventional imaging in a safe manner has been a challenge. The simplest solution has been to utilize larger treatment volumes or the use of fiducial markers, with the limitations that this entails (anesthesia, implantation technique, risk of migration, etc.). With an MR-LINAC, all this can be avoided.

#### 3.3.1. Liver Tumors

The liver is a frequent site of appearance for primary and metastatic tumors. With the introduction of SBRT as a therapeutic option, a direct relationship has been established between dose escalation and both local control [31] and OS [32]. This is also the case for cholangiocarcinoma, in which a BED > 80.5 Gy has been reported to show better three-year local control (78% vs. 45%) and OS (73% vs. 38%) compared to lower doses [33]. However, dose escalation has been difficult to implement in practice due to the radiological characteristics of the liver. Traditionally, gross tumor volume (GTV) delineation is performed aided by contrast-enhanced CT, positron emission tomography (PET), or MRI. Despite these tools, it is still a challenging task. MR-LINAC eases this problem with the use of “True-FISP” imaging for both simulation and treatment, allowing for better target characterization (the signal is hyperintense for hepatocarcinoma, isointense for cholangiocarcinoma, and hypointense for metastases) [34]. Another difficulty for liver SBRT is movement. It has been documented that the liver can move up to 2.4 mm in the lateral axis, 4.4 mm in the anteroposterior axis, and 14.7 mm in the craniocaudal axis [35,36]. 

MR-LINACs have integrated tumor-tracking systems that allow for treatment administration during breath-hold (gating), which is the most reproducible system [37,38] (Figure 2), although free-breathing treatment can also be performed. Online adaptation makes it possible to create a new treatment plan to always deliver a safe ablative dose [39], preserving healthy tissue in cases such as medial liver lesions close to the intestine and stomach [40].

Henke and colleagues at Washington University School of Medicine in 2018 evaluated 20 patients with unresectable primary liver and non-liver abdominal malignancies recruited to receive a median radiation dose of 50 Gy in five treatments. Within 1 year, the overall survival (OS) was 75%, and 15-month local control was 90%. Thus, SMART was clinically feasible, and ablative radiation doses were delivered with low rates of toxicity [41].

A Heidelberg University Hospital in Germany (Hoegen et al. [42]) treated 20 patients, 18 with liver metastasis and 2 HCC with ablative MR-guided SBRT, between January 2019 and February 2020. At 1 year, the LC was 88.1% and OS was 84.0%. Grade 2 toxicity occurred in 5.0% of the cases with no grade > 3 toxicity.

All of the studies mentioned here confirmed promising results, and future prospective clinical trials to validate their findings are warranted.

#### 3.3.2. Pancreatic Tumors

In this setting, multidisciplinary management with the aim of achieving R0 surgical resection is the main objective, but this is possible in only 15–20% of patients at diagnosis [43,44]. In patients with locally advanced pancreatic cancer (ALPC) treated with SBRT, the median OS was 17% with late grade 3–4 toxicity lower than 11% [45]. These results were achieved with conventional LINACs. However, the image quality of MRI improves the detection and delineation of the complete tumor extension compared to CT (Figure 3) [46]. The prospective SMART study evaluated the viability and safety of MRgRT in oligometastatic or non-resectable abdominal tumors using doses of 50 Gy in 5 consecutive fractions, with gating in MR-cine sequences in the exhalation phase and online ART in 91/97 fractions administered. Dosimetric advantages were obtained: the PTV dose could be increased up to 60 Gy in four fractions with better OAR protection [47], reaching a GTV coverage of 95%. The final PTV was GTV + 5 mm. In cases where the margin compromised the OAR tolerance, PTV coverage was sacrificed [48]. The main benefit of SMART was observed in cases where the distance from the GTV to the OAR was ≤3 mm [47]. Other studies have reported that BED10 > 70 Gy is associated with longer two-year survival compared to BED10 < 70 Gy (49% vs. 30% *p* = 0.03). In those with BED10 = 100 Gy (5 × 8–10 Gy), local control at one and two years was 94.6% and 83%, respectively. Grade 3 toxicity was 4.1% and 12.8%, respectively [49]. In short, dose escalation is justified for the treatment of these tumors, but the ideal fractionation and clinical scenario are yet to be determined. The application of five 6.6 Gy fractionations with adaptative SBRT has been described for long-term pain control [50].

In recurrent tumors, 30 Gy in five to six fractions [51] and 40 Gy in six fractions have been delivered after treatment with conventional fractionation, with no acute or late grade ≥ 3 toxicity [52]. In view of the variable approaches published, there is need for prospective randomized studies that validate the use of MR-LINAC in this scenario [53]. We are awaiting the results of the MOMENTUN trial (NCT04075305), a multi-institutional, international registry, and the phase II NCT03621644, that are currently recruiting patients with a dose prescription of 50 Gy in five fractions, studies that will help determine appropriate treatment doses in this type of tumor, oncologic outcomes, and quality of life.

#### 3.3.3. Rectal Cancer

For the last two decades, non-surgical approaches to treatment of this disease have been increasing in select patients with the use of total neoadjuvant therapy (TNT). MR-LINAC offers the possibility of intensifying the RT dose and, subsequently, increasing the rate of pathological response [54]. Passoni et al. [55] showed that a tumor boost of 54 Gy together with pelvic RT resulted in a 35% pathological complete response (pCR) [55]. If escalated to 60 Gy, pCR can be achieved in 78% of cases [56]. However, achieving these doses in clinical practice can be difficult due to uncertainties in organs at risk (bladder and rectal filling) and the impossibility of checking the treatment in real time. This requires increased treatment margins that lead to more toxicity. MRgRT paves the way for this dose escalation. Chiloiro et al. [57] treated 22 patients using MRIdian with doses of up to 55 Gy and reported 27% pCR, these results are no better than those described so far, although they should be evaluated under clinical trials with representative samples. Moreover, studies on MRI 3T have shown a significant tumor reduction during the first week of treatment, which further supports the use of ART [54]. According to some studies, the mean treatment time is 48 min [58]. Other studies focusing on ART have reported that the reduction in PTV (4 mm) translates to less gastrointestinal toxicity [59]. Functional imaging with diffusion-weighted imaging (DWI) and apparent diffusion coefficients (ADCs) can help adapt the RT plan, predict treatment response, and contribute to reducing treatment volumes [60].

### 3.4. Breast Cancer

Local recurrences in early-stage breast cancer occur in less of 5% of cases at ten years, according to recent studies [61,62,63]. This has caused a debate on whether treatment de-escalation is necessary [10]. In the case of RT, beginning with the results of phase 3 studies [61,63] there has been a trend towards accelerated partial breast irradiation (APBI) with external-beam radiotherapy, which has shown similar comparable results in terms of local recurrence, OS, and acute toxicity (Meattini et al. [63], NSABP B-39 [64], RAPID [65]). It must be noted that APBI allows for the treatment of just the surgical bed with margins of 1–1.5 cm for the clinical target volume (CTV) and a 1 cm expansion for the PTV [63,65,66]. Another scenario of great interest is neoadjuvant therapy [67], which can help achieve increased doses that, as seen in pancreatic tumors, are associated with better local control and, possibly, increased survival. However, this concept is not standard in breast tumors, which is why current studies are focusing on finding the optimal fractionation in order to avoid unnecessary irradiation of the whole breast [68]. It has been reported that most residual disease is located in the surgical bed, mainly within 1.5 cm of the location of the primary tumor [69].

MRI imaging provides better contrast between the target volume and the rest of the breast tissue. This helps achieve less irradiation to the chest wall, even delivering RT in a prone position. Moreover, the CTV and PTV margins could be reduced to obtain smaller treatment volumes and reduced toxicity [10]. Current publications with MR-LINAC in this setting are retrospective, mainly in the adjuvant setting [70,71,72,73,74] and less frequently in the neoadjuvant scenario [73,75]. The PTV margins are variable and are generated with a 5–10 mm expansion from the CTV, and with respiratory gating. Fractionations vary between studies from a single dose of 15, 18, or 20 Gy [75], to 5 fractions [71,76], or ≥10 fractions. These variations in dose do not seem to have consequences on the toxicity or cosmetic results [77]. An ongoing phase I clinical trial (NCT04849871), with the principal objective of which is the proportion of patients who are free of breast cancer in the treated breast (IBTR), compares single dose (20 Gy) APBI vs. APBI in five fractions (30 Gy) of adjuvant SBRT in any SBRT-ready LINAC such as MR-LINAC. Despite the lack of evidence and ongoing studies [73,75], it is feasible to deliver neoadjuvant RT with APBI due to the improved tumor visualization that contributes to a reduction in PTV margins.

### 3.5. Gynecological Tumors

MRI is part of the diagnosis of local tumors in this setting and is also useful for RT planning and brachytherapy (BT) delivery [78]. MRgRT has been used successfully in cervical, endometrial, and vaginal tumors, as well as in oligometastases from ovarian cancer [79] and some cases of tumor recurrences [80]. In the case of cervical cancer, an adequate identification of soft tissues is vital to achieve a precise contouring and margin reduction, so ongoing studies are focusing on hypofractionation as a boost for patients who are not candidates for BT [81]. As for inoperable patients with endometrial cancer due to old age, bad performance status, or comorbidities, they are usually managed with RT ± BT. Hypothetically, MR-LINAC could make it possible to deliver a boost to the uterus and/or positive nodes with better precision, but there is no published data on this setting. Another scenario yet to be explored but reported in case series is local recurrences in anatomical regions that are difficult to access with BT and those with a high risk of fistulization such as the rectovaginal septum and the rectal wall. These could be treated with less toxicity with MRgRT thanks to real-time image verification [81]. Through joint efforts and collaboration, clinical results to further explore the full potential of this innovative technology for the management of women with gynecologic cancer should be defined during the next decade.

### 3.6. Kidney Tumors

The gold standard for these tumors is surgery. In patients with comorbidities, there are ablative techniques such as radiofrequency or cryotherapy, or even active surveillance. However, these have limitations such as tumor size, proximity to blood vessels, or anesthesia [82]. Traditionally, RT has not been considered an option because these tumors are resistant to conventional fractionation. However, this has not been the case with the published studies on hypofractionation [83,84]. The study by the International Radiosurgery Oncology Consortium for Kidney (IROCK) showed that patients treated with SBRT in conventional LINACs have a local control of 98% and progression-free survival (PFS) of 65% at 4 years. Grade ≥ 3 toxicity was present in less than 2% of patients, and the impact on renal function was minimal [85].

Recently, at ASTRO 2023, Siva et al. presented data from the phase II study, FASTRACK II. A dose of 42 Gy was administered in three fractions and after a mean follow-up of 43 months the local control rate was 100%, with one patient only developing distant failure (freedom from distant failure = 99%). The cancer-specific survival was 100%, and the mean kidney function loss was −14.6 mL/min [86]. After these data, SBRT in this scenario is positioned as the treatment of choice in inoperable patients.

With the arrival of MRgRT, the movement of the kidney during treatment has been assessed in studies by Cusumano et al. [87,88]. They report that the kidney moves 4–9 mm in the craneocaudal axis and 2–3 mm anteroposterior, which is why real-time imaging is crucial to achieve better results with this technique. The first patient treated with MR-LINAC was reported by Rudra et al. [89], who administered 40 Gy in five fractions with 5 mm margins. No acute toxicity appeared, and the tumor response and renal function remained stable at least 6 months after SBRT. Tetar et al. [90] published the results of 36 patients with renal cell carcinoma treated with 40 Gy in five fractions. With a median follow-up of 16.4 months, the 1-year local control was 95.2% and OS was 91.2%. One patient suffered from grade 2 nausea and no other toxicities were reported. Following these results, there are ongoing prospective studies with SBRT: ANZUP (NCT02613819) and RADSTERM (NCT03811665). Its combination with immunotherapy is also being investigated (CYTOSHRINK NCT04090710). SBRT with MR-LINAC is appropriate and feasible in this group of patients, more so in the setting of oligometastatic patients receiving local treatment to both the primary tumor and distant metastases.

### 3.7. Central Nervous System Tumors

MRI is the gold standard for diagnosis and treatment response evaluation in brain tumors due to its excellent contrast of soft tissue (e.g., T1 and T2), and advanced techniques can be employed to assess physiology (diffusion, perfusion, spectroscopy). For this reason, MRgRT is very promising for the treatment of primary and metastatic brain tumors. Not so much because of movement, but due to the possibility of adapting interfraction anatomical changes and obtaining information about functional changes in the tumor such as cellularity, necrosis, and metabolism [91]. These systems can acquire weighted images in T1 and T2 to adapt the RT plan for each fraction. This helps maximize the benefit considering these dynamic changes, particularly in the setting of reduced CTV for gliomas and stereotactic fractionated RT (SFRT) to the surgical bed. However, there are still uncertainties in terms of target location after the configuration of translation-only MRI and intrafraction movement [92]. 

With MRgRT, it has been reported that the growth of glioblastoma (GBM) can be observed daily during RT [93]. Identifying these progressing patients during the first stages of treatment is key for applying aggressive second-line therapy or even dose escalation with multiparametric MRI (mpMRI) and image analysis. Early studies with MRgRT in this setting have been conducted with conventional fractionation [94,95,96], but ongoing phase II studies include moderate hypofractionation delivered in three weeks with reduced margins (Table 2).

There are currently several clinical trials investigating the use of MRgRT for the treatment of GBM. One is UNIty-Based MR-Linac Guided AdapTive RadiothErapy for High-GraDe Glioma: A Phase 2 Trial (UNITED) (NCT04726397) studying the use of a small margin (5 mm from GTV to CTV) with a daily-adaptive MRgRT approach for GBM using the Unity system with the objective of finding the “marginal” failure one year finished treatment (Table 2). The other one is UNITy-BasED MR-Linac Adaptive Simultaneous Integrated Hypofractionationed Boost Trial for High-Grade Glioma in the Elderly (UNITED2) (NCT05565521); this study is using a small-margin approach as well when applying treatments of 40 Gy in 15 fractions for elderly GBM, but adding a simultaneously integrated boost to 52.5 in 15 fractions to the GTV only (with a PTV expansion).

Stereotactic radiosurgery (SRS) is an effective option for brain metastases (BM). In large lesions (>2–2.5 cm), SFRT is preferred to reduce the risk of radionecrosis while preserving local control. The variations in size, geometry, and tumor location caused by the treatment can result in the delivery of an inadequate dose to the target and increased radiation to normal brain tissue, which can affect treatment efficacy [97]. An exploratory analysis of MRI-guided SFRT in patients with BM showed a significant dosimetric benefit in patients with perilesional edema or multiple lesions, achieved through daily reoptimization of the RT plan [98]. Ongoing prospective studies will provide more information in terms of tumor control and safety (Table 2). Changes in mpMRI could predict early response [99]. The anatomical adaptation could also be useful in single-fraction SRS in case of RM planning delays or for SFRT delivered for one week or more [100,101]. Post-operatory SRS in MR-LINAC using a post-surgical MRI is also feasible. However, the less favorable dose distribution of MR-LINAC compared to non-coplanar VMAT should be considered [100].

### 3.8. Miscellanea

#### 3.8.1. Heart Disease

Primary or secondary heart tumors are rare, with an incidence of 0.001–0.28% according to autopsies. Metastases from mesothelioma (48.5%), melanoma (27.8%), and lung adenocarcinoma (21%) are the most frequent, with the pericardium being the most affected region [102]. The resection of these lesions is a real challenge. MRI images in T1 and T2 can help characterize these tumors more precisely. Moreover, the implementation of CINE sequences makes it possible to follow movement during heartbeats, reducing ITV margins and eliminating the need for fiducials. In 2020, Sim et al. [103] reported on five patients treated with MR-LINAC to intracardiac lesions with a mean dose of 40–60 Gy and a 3 mm PTV margin with no ITV. The only acute side effect was grade 2 dyspnea. Furthermore, Gabani et al. [104] described the case of a patient with angiosarcoma treated with 30 Gy in five fractions and minimal acute toxicity.

Another emerging field in RT is the treatment of ventricular tachycardia with SBRT. The hypothesis is that ionizing radiation produces a homogenization of the scar tissue that is responsible for some of these tachycardias [105]. In 2017, Cuculich et al. [106] published their experience with five patients treated with single-fraction SBRT (25 Gy), with a 99.9% reduction in tachycardia episodes in the following 46 months. Data from the RAVENTA trial also support this approach in a further report on its safety [107].

#### 3.8.2. Nodal Metastases

Nodal SBRT in conventional LINACs has low toxicity. Cuccia et al. [108] conducted an observational study on 23 patients treated with 35 Gy in five fractions and comparing daily ART with MRI against the preplanned treatment in abdominopelvic nodal lesions. They reported a dose reduction in the bowel, with a maximum dose of 23.05 Gy (planned) vs. 20.5 Gy (adapted), and a mean dose of 14.4 Gy (planned) vs. 13 Gy (adapted) (*p* < 0.05 in both cases). Yoon et al. [109] reported one of the largest series of oligometastatic nodal lesions, treating 121 tumors with a mean dose of 40 Gy in five fractions. Of these, 13.9% were treated with ART when the dose to the OARs did not meet the constraints of the predicted plan. In terms of toxicity, grade 3 acute side effects were 0.9% and no grade 4–5 toxicity was reported. Late grade 3 toxicity was 5.2% and grade 4 was 2.1%. Two-year local control was 96% in those lesions receiving BED > 100 Gy.

#### 3.8.3. Adrenal Metastases

Evidence on this scenario is scarce to date. In one study, a series of 12 patients were treated with MRgRT with a median dose of 35–50 Gy in three to five fractions. The dose to the PTV was adapted daily when doses to OARs surpassed the predicted plan. This allowed for higher mean doses and PTV coverages (*p* = 0.04), which also helped reduce the maximum doses to OARs. At one year, local control was 100% and OS was 91.7%. No grade ≥ 2 toxicity was reported [109]. One of the advantages of MR-LINAC SBRT is the opportunity to increase the mean and high doses in the PTV while decreasing doses to OARs at the same time though daily ART. This characteristic is of great importance, as shown in the study by Buergy et al. [110], where a moderately higher median dose (BED10 > 73.2 Gy, 69.1 Gy for adenocarcinoma) decreased the risk of recurrence. These data were corroborated by Hoegen et al. [111], who analyzed 22 patients treated with adrenal SBRT guided by MRI. In the predicted plan, the PTV coverage was 99.4% and the OAR restrictions were as follows: intestine: 32.9%, stomach: 32.8%, duodenum: 10%, kidneys: 10%. Whereas using adapted plans improved the PTV coverage from 82.7 ± 8.1% to 90.6 ± 4.9% (*p* < 0.001) (Figure 4).

#### 3.8.4. Spinal Metastases

One of the challenges in spinal SBRT is the precise delineation of the spinal cord as a critical organ. Low coverage in areas adjacent to the spinal cord can result in tumor recurrences; therefore, good image control could help improve this coverage, which would translate into better local control. Oztek et al. [112] conducted a descriptive study with the aim of evaluating the inherent movement of the spinal cord through an MRI dynamic sequence known as dynamic balanced fast field echo (BFFE). In this study, 21 patients were treated with spinal SBRT aided by image fusion with this dynamic sequence. In 62% of cases, the dose to the spinal cord exceeded between 0.6% and 13.8% when no PRV was defined. In contrast, the use of a 2 mm PRV kept the treatment inside the planned dose. MRI-guided SBRT could potentially avoid this uncertainty. In 2019, Redler et al. studied the dosimetric feasibility of eight cases replanned for MR-LINAC using CT and MRI images from CT and VMAT-based treatments, and obtained positive results in terms of a low dose to the spinal cord and improved PTV coverage [113].

#### 3.8.5. Head and Neck

The use of MR-LINAC in head and neck cancer has the potential for treatment personalization and may offer an opportunity to study anatomical changes and response during treatment with the collection of serial MRIs, including functional sequences. This amount of imaging data may also be utilized to develop image-based predictive outcome models. Nevertheless, MRgRT to date presents some limitations, including the inability to support volumetric arc therapy.

Overall, this innovative MRgRT has the potential to improve patient outcomes while minimizing the risk of toxicity by providing precise, high-quality imaging and daily-adapted radiation delivery. Data on the use of MR-LINACs for head and neck cancers is, however, scarce and most of the initial experiences are under evaluation [114].

## 4. Future Directions

The treatment duration in MR-LINAC has been optimized with the added possibility of parallel workflows for ART during each fraction. One debated topic in this optimization has been the requirement of a simulation CT for the calculation of dose distribution, given that Hounsfield units cannot be obtained from MRI. Several solutions have been proposed that generate synthetic CT from MRI images: bulk density, atlas-based, and machine learning methods, with the latter being the most promising [10]. Another noteworthy aspect is radiomics, which is defined as the application of quantitative imaging analysis to integrate clinical data and improve decision-making [115]. This can be useful to predict tumor response based not only on the morphological image, but also incorporating functional data for survival models. All these advances will help in the optimization of these treatments. A feasible integration of this technique into radiation oncology departments and the results of the ongoing clinical trials may help MR-LINAC to become the gold standard in many situations.

Another aspect includes the fact that we currently have two commercialized MR-LINAC systems, which raises their costs compared to other LINACs, as well as the fact that MR-LINAC systems treat fewer patients per day but with more complex treatments. In the future, with the integration of other systems into the market, they will surely become more accessible and essential parts of a radiation oncology department.

## 5. Conclusions

MRgRT is a new technology that changes the treatment paradigm of radiation therapy by offering the possibility of increasing the quality of daily re-contouring, adapted planning, and real-time tumor tracking. These advances, that have improved treatment precision, have motivated the reduction in planning margins with a decrease in side effects for patients. The technological advances of MRgRT have also facilitated safe dose escalation in tumors that had been historically limited by juxtaposed critical OARs, such as for pancreatic cancer, in which dose is related to local control. In contrast, these technologies are expensive and for now require a long time to be spent with each patient and more human resources than in conventional treatments, but hopefully with positive results. In the coming years, more tumor types will be treated with MR-LINAC, with adapted and easier workflows that translate into shorter treatment durations. It is inevitable that MRgRT will revolutionize radiation oncology in decades to come. 

## Figures and Tables

**Figure 1 cancers-16-00270-f001:**
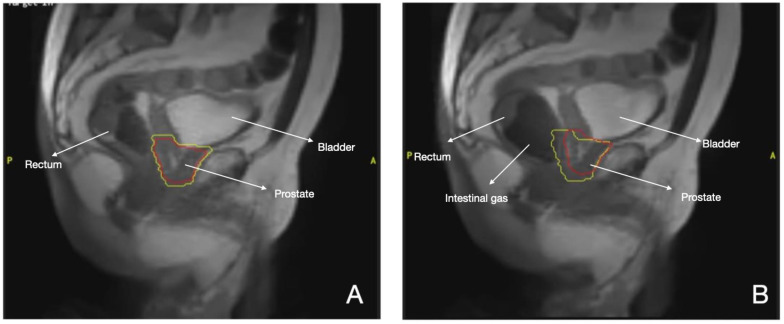
MR-LINAC prostate cancer treatment. Red line: Prostate. Yellow line: Margins of 2mm (**A**) Gating during SBRT, (**B**) Displacement of gland by intestinal gas.

**Figure 2 cancers-16-00270-f002:**
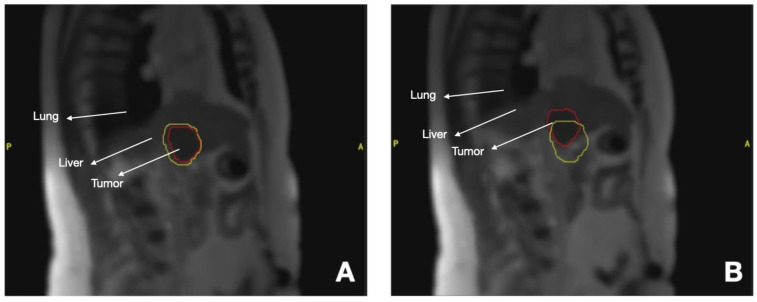
MR-LINAC Liver metastases treatment. Red line: Liver Metastases. Yellow line: Margins of 3 mm (**A**) Gating during SBRT, (**B**) Displacement of tumor during respiration.

**Figure 3 cancers-16-00270-f003:**
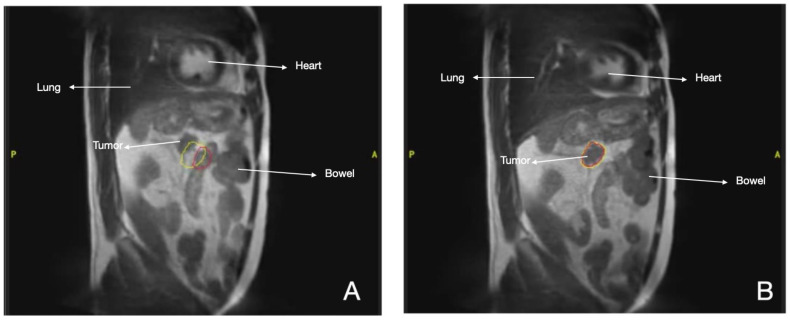
MR-LINAC Pancreatic cancer treatment. Red line: Pancreatic tumor. Yellow line: Margins of 3 mm (**A**) Gating during SBRT, (**B**) Displacement of tumor during respiration.

**Figure 4 cancers-16-00270-f004:**
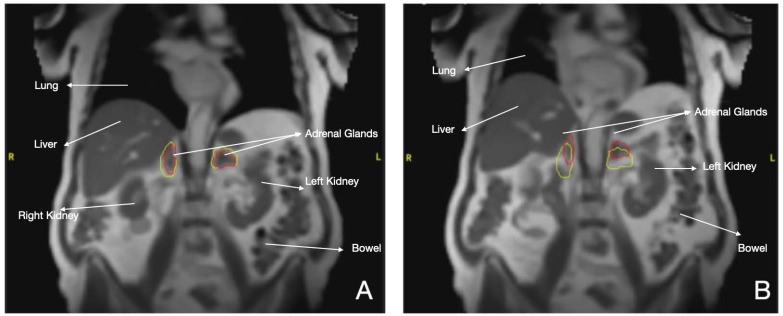
MR-LINAC of two metastases of lung cancer in adrenal glands. Red line: Adrenal glands with tumor. Yellow line: Margins of 3 mm (**A**) Gating during SBRT, (**B**) Displacement of two tumors during respiration.

**Table 1 cancers-16-00270-t001:** MRIdian vs. Unity comparison.

	MRIdian	Unity
Magnet design	Split magnet	Single magnet design
Magnet	0.35 T *	1.5 T *
Intrafraction management	Automatic	Non-automatic
Image	TRUFI **, T1, T2	Variety of image sequences
Tracking	Real-time tracking	Real-time tracking
Gating	Automatic	Non-automatic
Gantry	Maximum rotation speed 0.5 rpm ***	Maximum rotation speed 6 rpm ***

*: Tesla. **: True fast imaging with steady-state free precession. ***: revolutions per minute.

**Table 2 cancers-16-00270-t002:** Ongoing clinical trials with MRI-LINAC.

Study Name	Tumor Type	Study Type	Intervention	Objective
Dose de-escalation in prostate radiotherapy using the MRL (DESTINATION) NCT05709496	Prostate cancer	Phase II	Radiotherapy to the prostate which will be given in 30 Gy in 5 fractions to the whole prostate and 45 Gy in 5 fractions to the dominant lesion.	The goal of this feasibility study is to learn about dose de-escalation in the treatment of men with intermediate-risk prostate cancer.
Randomized trial of five or two MRI-Guided adaptive radiotherapy treatments for prostate cancer (FORT) NCT04984343	Prostate cancer	Phase II	Patients will receive 25 Gy in two radiotherapy treatments vs. 37.5 Gy in 5 fractions.	To demonstrate that two treatments of radiotherapy does not significantly increase patient-reported gastrointestinal (GI) and genitourinary (GU) symptoms compared to five treatments of radiotherapy 2 years after treatment completion.
MR-LINAC Guided Ultra-hypofractionated RT for Prostate Cancer (NCT05183074)	Prostate cancer	Phase II	Ultra-hypofractionated radiotherapy in patients with low-, intermediate, and high-risk prostate cancer.	To investigate the tolerability of MR-LINAC based stereotactic ablative radiotherapy (MRL-SBRT) for patients with localized prostate cancer.
Magnetic Resonance Guided Adaptive Stereotactic Body Radiotherapy for Lung Tumors in Ultracentral Location (MAGELLAN) NCT04925583	Lung cancer	Phase I	Patients will receive 50 Gy in 5 fractions; 55 Gy in 5 fractions; 60 Gy in 5 fractions; or 65 Gy in 5 fractions.	To identify the maximum tolerated dose (MTD) of MRI-guided SBRT to ultracentral lung tumors.
Magnetic Resonance-Guided Hypofractionated Adaptive Radiation Therapy With Concurrent Chemotherapy and Consolidation Durvalumab for Inoperable Stage IIB, IIIA, and Select IIIB and IIIC Non-small Cell Lung Cancer NCT03916419	Lung cancer	Phase II	Patients will receive 60 Gy in 15 fractions.	Safety of hypofractionated MRI-guided adaptive radiotherapy.
Magnetic Resonance-guided Adaptive Stereotactic Body Radiotherapy for Hepatic Metastases (MAESTRO) NCT05027711	Liver metastases	Phase I	SBRT with MRI-LINAC BED ≥ 100 Gy with ITV vs. SBRT with LINAC ITV ≥ 100 Gy vs. SBRT with MRI-LINAC to the highest possible dose to the ITV.	Treatment-related toxicity.
Locally Advanced Pancreatic Cancer Treated With ABLAtivE Stereotactic MRI-guided Adaptive Radiation Therapy (LAP-ABLATE) NCT05585554	Pancreatic cancer	Phase I–II	Induction chemotherapy + MR-LINAC 50 Gy vs. induction chemotherapy alone.	To demonstrate superior 2-year overall survival from date of randomization in ablative MRIdian SMART versus no ablative MRIdian SMART in locally advanced pancreatic cancer patients without disease progression after induction chemotherapy.
Chemotherapy Combined With High-dose Radiotherapy for Low Rectal Cancer Using MR Guided Linear Accelerator NCT05338866	Rectal cancer	Phase II	CT-RT in LINAC (50 Gy in 25 fractions) + Boost in MR-LINAC 16–20 Gy in 8–10 fractions vs. CT-RT in LINAC (50 Gy in 25 fractions) + Boost in MR-LINAC 30 Gy in 15 fractions.	Three-year progression-free survival rate.
MRI-Guided Radiation Therapy for the Treatment of Early-Stage Kidney Cancer, the MRI-MARK Trial NCT04580836	Kidney cancer	Phase II	Treatment (MRI-guided SBRT).	To evaluate local control following magnetic resonance imaging (MRI)-guided stereotactic body radiation therapy (SBRT) for primary kidney cancer, as defined by no growth by imaging at 24 months following SBRT.
UNITED (UNIty-Based MR-Linac Guided AdapTive RadiothErapy for High GraDe Glioma: A Phase 2 Trial (UNITED) NCT04726397	Glioblastoma multiforme	Phase II	Radiation: adaptative radiotherapy with reduced margins (CTV 5 mm)	Tumor recurrence detected by imaging at the edge of the radiation volume.
UNITy-BasED MR-Linac Adaptive Simultaneous Integrated Hypofractionationed Boost Trial for High Grade Glioma in the Elderly (UNITED2) NCT05565521	Glioblastoma multiforme, adult IDH-mutant glioblastoma	Phase II	Radiation: dose escalation + adaptative radiotherapy with reduced margins.	Progression-free survival at 6 months after chemoradiation.
A Prospective, Phase II Study of MR-Linac Guided Adaptive Fractionated Stereotactic Radiotherapy for Brain Metastases From Non-small Cell Lung Cancer NCT04946019	1–10 BM non-small cell lung cancer	Phase II	Combined product: FSRT guided by Unity-based MR-LINAC FSRT (30 Gy in five fractions).	Intracranial progression-free survival at 1 year.
A Master Protocol of Stereotactic Magnetic Resonance Guided Adaptive Radiation Therapy (SMART) NCT04115254	All/multiple sites (including BM)	Phase I–II	MRI-guided LINAC.	Phase I: radiation will be administered in an MRI-LINAC. Phase II: Improvement in local control.

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
