# Peer review of "MR-LINAC, a New Partner in Radiation Oncology: Current Landscape"

_cancers, 2024, doi:10.3390/cancers16020270_

Round 1
Reviewer 1 Report
Comments and Suggestions for Authors
General Comments
1 The authors have provided an excellent comprehensive review on the development, clinical studies, and potential clinical benefits of MRgRT in treatment of a wide range of tumor types. The information provided should be of value to the radiation oncology community, particularly those who are justifying for their request for acquisition of the technology.
2 The manuscript is recommended for acceptance for publication subject to satisfactory addressing by the authors the issues raised in the specific comments listed below.
Specific Comments
Lines 28-29: “risk that can move” appears to be an inappropriate phrase in the context of this sentence.
Lines 29-30: “conventional radiation therapy” should be more specifically defined, e.g. X-ray-based IGRT, because MRgRT is also regarded as conventional RT in the particle therapy community.
Line 34: Spelling error in “lineal accelerators”.
Line 57: The original paper published in 1997 by D. Yan introducing the concept of ART should be referred to.
Line 65: Spelling error in “beling”.
Line 66: “magnetic resonance-based’ should be “magnetic resonance imaging-based”.
Line 74 and more: Spelling error in “imrpofved”. Numerous other typing and grammatical errors are found in various parts of the manuscript.
Table 1: The features/parameters selected by the authors in the comparison is not comprehensive. The authors should explain the reasons for selecting such features for comparison.
Lines 211 and 212: To be consistent with the terminologies introduced earlier in the text, MRI group and CT group should be used instead of MRI and CT.
Line 229: A reference to the SHORTER trial paper by Marciscano et al should be given.
Lines 264, 363, 398, 441, 454 477, 497, 507 & 511: The term MRIgRT is used while MRgRT is used elsewhere, including those in Abstract. The authors should be consistent with the terminology used.
Line 277: SBRT guided by MR-LINAC is an incorrect term. The treatment is guided by MRI not by the LINAC.
Line 382-384: It is not clear what the justifications for the proposed prospective randomized studies are.
Lines 385-387: A short description on these two MOMENTUM studies or, preferably a reference to each of them should be provided.
Lines 398-399: The outcome of the 55Gy MRIdian MR-LINAC treatments is poor as compared with other treatments described earlier in the text. This undermines the value of MR-LINAC and contradicts with the conclusions stated in the paper.
Comments on the Quality of English LanguageApart from some typing and grammatical errors, the quality of English language is satisfactory in general.
Author Response
Dear Reviewer, thank you for your suggestion about our paper, we described all changed made:
- Lines 28-29: “risk that can move” appears to be an inappropriate phrase in the context of this sentence: Changed.
- Lines 29-30: “conventional radiation therapy” should be more specifically defined, e.g. X-ray-based IGRT, because MRgRT is also regarded as conventional RT in the particle therapy community: Changed.
- Line 34: Spelling error in “lineal accelerators”: Corrected.
- Line 57: The original paper published in 1997 by D. Yan introducing the concept of ART should be referred to: Added and referenced.
- Line 65: Spelling error in “beling”: Changed.
- Line 66: “magnetic resonance-based’ should be “magnetic resonance imaging-based”: Changed.
- Line 74 and more: Spelling error in “imrpofved”. Numerous other typing and grammatical errors are found in various parts of the manuscript: Changed.
- Table 1: The features/parameters selected by the authors in the comparison is not comprehensive. The authors should explain the reasons for selecting such features for comparison: We use the technical parameters more important between the two commercialized MRI-LINACs.
- Lines 211 and 212: To be consistent with the terminologies introduced earlier in the text, MRI group and CT group should be used instead of MRI and CT: Changed and adjusted.
- Line 229: A reference to the SHORTER trial paper by Marciscano et al should be given: Added.
- Lines 264, 363, 398, 441, 454 477, 497, 507 & 511: The term MRIgRT is used while MRgRT is used elsewhere, including those in Abstract. The authors should be consistent with the terminology used: Changed.
- Line 277: SBRT guided by MR-LINAC is an incorrect term. The treatment is guided by MRI not by the LINAC: Changed.
- Line 382-384: It is not clear what the justifications for the proposed prospective randomized studies are: We modified the paragraph and describe the intention of MOMENTUM trial.
- Lines 385-387: A short description on these two MOMENTUM studies or, preferably a reference to each of them should be provided: We modified the paragraph and describe the intention of MOMENTUM trial.
- Lines 398-399: The outcome of the 55Gy MRIdian MR-LINAC treatments is poor as compared with other treatments described earlier in the text. This undermines the value of MR-LINAC and contradicts with the conclusions stated in the paper: We added few words and describe the importance of good clinical trials in this scenery and the small sample of the study referenced.
Thank you for all support and suggestion
Kind regards
Abrahams Ocanto
Reviewer 2 Report
Comments and Suggestions for Authors
This paper provides a current overview of the use of MR-LINAC technology in the treatment of solid tumors via radiotherapy. The authors comprehensively discuss the advantages of this technique, such as superior visualization and the possibility of online adaptive radiotherapy. The authors also presented selected clinical data regarding the efficacy of MR-LINAC for the treatment of various cancers.
However, some imperfections of the manuscript should be noted:
1. There is no extensive presentation of long-term results concerning patient survival and the toxicity of particular treatment regimens. Most of the data focused on local control and acute toxicity. Long-term data are lacking.
2. The paper concentrated mainly on the description of the technique itself, while too little space was devoted to clinical achievements. A collection of hard data confirming the actual benefits for patients is necessary.
3. There is no comparison of MR-LINAC with other advanced radiotherapy technologies, such as VMAT with IGRT or proton therapy. Comparative analysis would be extremely helpful for readers.
4. The article has a mainly descriptive character, with few critical analyses. The authors uncritically accept the advantages of the new technique, omitting potential disadvantages and challenges.
In summary, although the paper provides a useful overview of MR-LINAC technology, further prospective studies verifying the actual long-term clinical benefits of this technique are necessary. Comparative analyses with other techniques would be advisable. The authors’ approach is slightly too enthusiastic and uncritical. Nevertheless, the article constitutes an interesting introduction into this innovative technology.
Author Response
Dear reviewer,
Thank you very much for reading our article, for your suggestions and comments, they are very helpful.
Regarding the suggested changes:
1: To date we only have studies with limited results, there is no published phase 3 study evaluating this parameter nor descriptive studies, that is why they are not included in our review.
2: The most information available corresponds to the dosimetric and technical advantages of this type of linear accelerators. We have therefore described these characteristics exhaustively.
3: Indeed, we include data from the MIRAGE study, the only phase III study comparing MR-LINAC treatment with conventional LINAC. Assessed in prostate cancer. With respect to proton therapy, in view of the fact that it is another particle and with more evidence in pediatric treatments where we do not have studies on MR-LINAC, we did not consider the comparison pertinent. MR-LINACs have photons, the same energy as conventional LINACs, therefore no dosimetric advantages are gained, except in the reduction of margins volumes, which is what has been widely commented on.
4: Indeed, we have added a few lines in the conclusion, highlighting the greater consumption of human resources and time in each treatment, which does not make it suitable for treating numerous patients in a working day.
We hope to be able to clarify some of your doubts, we are open to new corrections and indications to improve our article.
Best regards
Reviewer 3 Report
Comments and Suggestions for Authors
MRI-LINAC is a relatively recent promising technique in radiation oncology. The present critical review is timely and useful.
Minor comments:
One. MRI figures (1, 2, 3 and 4) should be labeled (organs at risk etc.) so as to be readable by the non-expert.
Two. Most studies admittedly examine safety with toxicity as the mean endpoint. Clinical trials evaluating cancer outcome for MRI-LINAC as compared to other techniques are exceptional yet (examples are Prostate : line 239, biochemical RFS; Lung cancer: Line 277, local control; line 385, pancreatic cancer; line 433, breast cancer; line 514, glioblastoma; see also Table 2). Description of such trials merits more detail about the study method (endpoint? Retrospective? randomized, etc.)?
Three. Financial aspects: could the authors add some information of the increased costs of MRI-LINAC, if any?
Author Response
Dear reviewer,
Thank you for your comments and suggestions, we have considered each of them and adjusted in the new manuscript.
We list the changes below:
One. MRI figures (1, 2, 3 and 4) should be labeled (organs at risk etc.) so as to be readable by the non-expert: We attached the figures labeled.
Two. Most studies admittedly examine safety with toxicity as the mean endpoint. Clinical trials evaluating cancer outcome for MRI-LINAC as compared to other techniques are exceptional yet (examples are Prostate : line 239, biochemical RFS; Lung cancer: Line 277, local control; line 385, pancreatic cancer; line 433, breast cancer; line 514, glioblastoma; see also Table 2). Description of such trials merits more detail about the study method (endpoint? Retrospective? randomized, etc.)?: We added the studies recommended, highlighting the most important aspect of each one.
Three. Financial aspects: could the authors add some information of the increased costs of MRI-LINAC, if any?: We added a few words in the future directions highlighting the increased cost of MR-LINAC system and the few patients that could be treated per day.
Kind regards
Abrahams Ocanto
